# Colorectal Cancer and Nutrition

**DOI:** 10.3390/nu11010164

**Published:** 2019-01-14

**Authors:** Kannan Thanikachalam, Gazala Khan

**Affiliations:** Department of Hematology/Oncology, Henry Ford Health System, Detroit, MI 48202, USA; gkhan1@hfhs.org

**Keywords:** Colorectal Cancer, Nutrition, Adjunctive therapy

## Abstract

Colorectal Cancer is the third most common cancer diagnosed in the US. While the incidence and the mortality rate of colorectal cancer has decreased due to effective cancer screening measures, there has been an increase in number of young patients diagnosed in colon cancer due to unclear reasons at this point of time. While environmental and genetic factors play a major role in the pathogenesis of colon cancer, extensive research has suggested that nutrition may play both a causal and protective role in the development of colon cancer. In this review article, we aim to provide a review of factors that play a major role in development of colorectal cancer.

## 1. Introduction

Colorectal cancer (CRC) is the third most common cancer diagnosed in both males and females in the US, (after prostate cancer in men, breast cancer in women and lung cancer). Approximately 41% of all colorectal cancers occur in the proximal colon, with approximately 22% involving distal colon and 28% involving rectum [1]. However, there can be potential differences in the site of origin depending on age and gender. Though the incidence and the mortality rate of colorectal cancer has decreased due to effective cancer screening measures, it has been projected that there will be 140,250 new cases of colorectal cancer in 2018, with an estimated 50,630 people dying of this disease [1,2]. Environmental and genetic factors play a major role in the pathogenesis of colon cancer. The role of nutrition in colon cancer has been studied extensively, explaining causal and protective role in the development of colon cancer. In this review article, we aim to focus on the role of nutrition in colorectal cancer.

## 2. Epidemiology

The incidence of colorectal cancer differs in each country. Several factors are thought to contribute to this variability in incidence. Specifically, amongst various factors socioeconomic status, with low socio-economic status being associated with an increased risk of development of colorectal cancer. In the US, the incidence of colorectal cancer has decreased from 60.5 per 100,000 people in 1976 to 46.4 per 100,000 people in 2005 [1]. The incidence of colorectal cancer had decreased at a rate of approximately 3% per year between 2003 and 2012 [2,3]. The incidence of colorectal cancer in the US in 2017 was 135,430 and deaths from colorectal cancer was 50,260 [3]. While the incidence of colorectal cancer has decreased overall, the incidence in men and women less than 50 years of age, has increased by 2%. It has been projected that incidence rates for colon and rectal cancers may increase by 90.0% and 124.2%, respectively, for patients between the ages 20 to 34 years by 2030 [4]. It is thought that about 35% of these young adult colorectal cancers are associated with hereditary colorectal cancer syndromes and the reason for the increase in the incidence is currently unknown.

Due to effective screening measures, early interventions, better treatment options, mortality from colorectal cancer has decreased by about 35% from 1990 to 2007 and is currently down by about 50% from peak mortality rates [5]. However, it is also to be noted that the decrease in overall mortality from colorectal cancer might have masked the mortality rate for young adult patients with colorectal cancer. Based on data from SEER database, while the mortality rate for young adults (20–54 years of age, white) decreased from 6.3 per 100,000 in 1970 to 3.9 in 2004, it increased to 4.3 per 100,000 in 2014.

## 3. Risk Factors

Age, genetic and environmental factors play a major role in the development of colorectal cancer. Hereditary colorectal cancer syndromes include Lynch Syndrome (Hereditary nonpolyposis colorectal cancer), Familial adenomatous polyposis (FAP), MUTYH-associated polyposis (MAP). Lynch syndrome and Familial adenomatous polyposis contribute to a vast majority of hereditary colorectal cancer syndrome, which account for only about 5% of entire colorectal cancer incidence [6]. The presence of family history of colon cancer in first degree relatives, even in the absence of the above hereditary colon cancer syndromes, increase the risk of development of colorectal cancer in about 20% of cases. The risk increases over twofold, when compared to the general population, with a history of colorectal cancer in first degree relatives.

Other well-known associations with colorectal cancer include African American ethnicity, male sex, Inflammatory Bowel disease—Ulcerative colitis more often than Crohn’s disease, Obesity, sedentary lifestyle, red meat (Table 1) and processed meat, tobacco use, alcohol use, history of abdominal radiation, acromegaly, renal transplant with use of immunosuppressive medications, Diabetes mellitus and insulin resistance, androgen deprivation therapy, cholecystectomy, coronary artery disease and ureterocolic anastomosis [6]. 

Protective factors that have been associated with a decrease in the incidence of CRC include regular physical activity, diet rich in fruits and vegetables, high fiber diet, folate rich diet, Calcium, Dairy products, Vitamin D, Vitamin B6, magnesium intake, fish consumption, garlic, regular use of Aspirin, Non-Steroidal Anti-Inflammatory Drugs (NSAIDS). 

A recent meta-analysis by Shivappa et al. demonstrated increased risk in the incidence of CRC with certain foods by the use of Dietary Inflammatory Index of food [13]. A higher DII score correlated with the pro-inflammatory potential, thereby increasing the risk of CRC, where as a lower DII score correlated with anti-inflammatory potential, thereby reducing the risk of CRC. Anti-inflammatory food components included fiber, alcohol, monounsaturated fatty acids, polyunsaturated fatty acids, omega 3, omega 6, niacin, thiamin, riboflavin, vitamin B6, B12, zinc, magnesium, selenium, vitamin A, vitamin C, vitamin D, vitamin E, folic acid, beta carotene, anthocyanidins, flavan-3-ols, flavonols, flavanones, flavones, isoflavones, garlic, ginger, onions, thyme, oregano, saffron, turmeric, rosemary, eugenol, caffeine and tea. Pro-inflammatory food components included energy, carbohydrates, proteins, total fat, trans fat, cholesterol, vitamin B12, saturated fatty acids and iron. 

Several studies have shown a causal role between alcohol consumption and incidence of colorectal cancer (Table 2). Meta-analysis of prospective studies have shown a modest positive association between heavy alcohol use (>50 g/day) and mortality associated with colorectal cancer [14]. This association was stronger in Asian population than in the white population, likely as a result of genetic factors such as alcohol metabolism, dietary factors such as folate intake and body composition. The association of alcohol drinking and risk of mortality from colorectal cancer was similar at different anatomic sites, such as colon and rectum. A prospective study of Carriers of Mismatch repair deficiency also revealed a positive association with alcohol consumption (>28 g/day or 2 drinks per day) and colon cancer [15]. It has been suggested that acetaldehyde, a metabolite of ethanol, is carcinogenic by affecting DNA synthesis, repair, alteration of structure and function of glutathione and increase in colonic mucosal proliferation. 

Multiple observational studies have shown an association between obesity and risk of colorectal cancer (20–30% per 5 kg/m^2^ increase in men and ~10% per 5 kg/m^2^ increase in women) [19]. A Mendelian Randomization Study showed a stronger association with obesity and colorectal cancer in women than in men [19].

Epidemiological studies have suggested a protective role for Vitamin D in the development of colorectal cancer. Both free and total 25-hydroxyvitamin D were shown to be inversely associated with colorectal cancer. A 10 ng increase in circulating Vitamin D level was associated with a 26% decreased risk of developing colorectal cancer [20]. It has been suggested that vitamin D receptor potentially mediates the protective effect of Vitamin D. A recent study, where data was pooled from 17 cohorts, involving 5706 colorectal cancer patients and 7107 control participants, revealed that higher circulating Vitamin D levels, lead to statistically significant decreased colorectal cancer risk in women and non-statistically significant reduction in men [21] (Table 3). The ideal Vitamin D concentration required for colorectal cancer risk reduction was also suggested to be 75–100nmol/L [21]. The Institute of Medicine’s recommendations for Vitamin D supplementation are based on data for bone health.

The role of Statins, Angiotensin II inhibition, Postmenopausal hormone therapy in women in the risk of development of colorectal cancer remains controversial. European Prospective Investigation into Cancer and Nutrition (EPIC)Cohort study evaluating pre-diagnostic intake of dairy products, dietary calcium and colorectal cancer survival did not show an association with reported pre-diagnostic intake of dairy products (milk, yogurt and cheese), dietary calcium and risk of colorectal cancer–specific death or that of all-cause death [23]. While several observational studies have suggested a protective role of calcium, at least two randomized controlled clinical trials failed to show a protective benefit [24,25] (Table 4).

### 3.1. Role of Gut Microbiome in Pathogenesis

Since Burkitt first linked that high fiber intake could be protective against the development of colorectal carcinoma [28], several studies have focused on the role of gut microbiota in the pathogenesis of colorectal cancer. As a result of investigations, it has been suggested that pro-inflammatory bacteria such as Fusobacterium nucleatum simulates local inflammatory response and suppresses immune reactions, in addition to activating the Wingless/Integrated-1 (WNT) signaling pathway [29]. The pro-inflammatory state of Fusobacterium nucleatum has also shown to contribute to epigenetic silencing of MMR Protein—MLH1, which may lead to Microsatellite Instable CRC [30]. Investigational studies comparing gut microbiota in African American patients and Caucasian patients who did not have any active malignancy showed higher incidence of pro-inflammatory bacteria in African American population and higher incidence of protective bacteria in Caucasian population [31]. Though there is evidence of the role of gut microbiota in the pathogenesis of CRC, the impact of lifestyle changes on anticancer immune response remains unclear. 

### 3.2. Fiber and Its Effect

Several epidemiological studies and two randomized control studies have been performed to assess the role of the fiber in pathogenesis of CRC (Table 5). While multiple epidemiological studies have suggested an inverse relationship between colorectal adenomas and carcinomas to fiber intake [32,33,34,35], two prospective studies found no relationship between fiber intake and CRC [12,36]. Three randomized controlled clinical trials also failed to demonstrate the protective benefit of fiber against CRC [37,38,39]. Though a pooled analysis of multiple prospective studies demonstrated an inverse relationship between dietary fiber intake and colorectal cancer, after it was adjusted for other dietary risk factors, it failed to show a benefit [40]. A meta-analysis showed that cereal fiber and whole grains had the most benefit against CRC [41]. A prospective cohort study demonstrated that a prudent diet rich in whole grains and dietary fiber were associated with a decreased risk for Fusobacterium nucleatum positive CRC but not Fusobacterium nucleatum negative CRC [42]. Role of resistant starch in protection against colorectal cancer remains an area of investigation. Resistant starch, on fermentation, generates Short Chain Fatty Acids—Butyrate among one of them. It has been shown that butyrate may have anti-neoplastic properties in colon by modulating immune response by suppressing histone deacetylation, thereby resulting in further protection against CRC. However, a randomized clinical trial failed to show any benefit with administration of resistant starch—Novolose 30 g daily in the development of adenoma or carcinomas in patients with Lynch Syndrome [43,44]. A recent study demonstrated that, in healthy individuals, fiber intake alters gut microbiota [45]. In a recent prospective study involving 1575 patients with Stage I to III colorectal cancer, high fiber intake—especially from cereals was associated with a low colorectal cancer specific mortality and over all mortality [46]. 

## 4. Clinical Presentation

Early stage colorectal cancers are commonly diagnosed by routine colonoscopies (both screening and surveillance). The US Preventive Services Task Force (USPSTF) recommends starting screening colonoscopy at the age of 50 years, followed by every ten years later as long as previous colonoscopy was normal [49]. American Cancer Society recommends starting colonoscopy at the age of 45 [50]. Common symptoms on presentation include change in bowel habits, hematochezia from rectal bleeding, Iron deficiency anemia, abdominal pain, loss of weight and loss of appetite. Approximately 20% of individuals who are diagnosed with CRC have metastatic disease on presentation. Metastasis occurs by lymphatic spread, hematogenous spread, contiguous or transperitoneal spread. Most common sites of metastases from CRC include regional lymph nodes, liver, lung and peritoneum. Depending on the site of metastases, symptoms may include abdominal pain, perforation and abscess due to direct extension, jaundice and right upper quadrant pain (Liver), supraclavicular lymphadenopathy, periumbilical nodules, Dyspnea (Lungs). Regardless of the stage of the cancer, intestinal obstruction and/or perforation signifies a poor prognosis. 

## 5. Diagnosis

Colonoscopy remains the study of choice to diagnose colorectal cancer. Prior to any treatment, CT imaging of chest, abdomen and pelvis with contrast is needed for staging patient’s CRC. Staging is commonly done by using Primary Tumor size (T), regional lymph Node (N) and distant Metastasis (M) - TNM classification system. Though tumor marker levels such as Carcinoembryonic antigen (CEA) levels can be elevated in colorectal cancer, it is not diagnostic of CRC. CEA levels are rather used as a tool to monitor in the post-treatment follow up and for surveillance. The most common lab parameter that is abnormal in patients with liver metastases is elevated alkaline phosphatase level. In patients with background liver disease, Magnetic Resonance Imaging (MRI) liver with contrast may add more accuracy in diagnosing liver metastases. 

## 6. Treatment Overview

At the time of diagnosis of colon cancer, approximately 80% are localized, whereas 20% have metastasized into distant sites. Surgical resection remains the only curative option for both colon and rectal cancers, that are loco-regional. Clinically occult micro metastasis can potentially occur at the site of surgery and adjuvant chemotherapy helps in eradicating micro-metastases. For locally advanced colorectal cancers, neoadjuvant chemotherapy is sometimes indicated. For colon cancers that involves lymph nodes (Stage III and above) or distant sites, chemotherapy is indicated either in the adjuvant setting or palliative setting. Chemoradiation is often required for locally advanced rectal cancer after surgical removal. Immunotherapy with Pembrolizumab is also an option for metastatic colorectal cancers that are microsatellite unstable. 

## 7. Adjunctive Therapy after Surgical Resection

Several factors such as diet changes, increased physical activity, Aspirin and NSAID use, Vitamin D status, coffee intake have been shown to be beneficial in the post treatment setting. While there are no randomized clinical trials that evaluated the role of diet after surgery, at least two studies showed that patients who were on a diet with increased processed meat, red meat, sweets, refined grains had increased recurrence rates and decreased disease free survival (DFS) rates [51,52,53]. A diet high in glycemic index was also associated with a decreased DFS among obese and overweight patients [53,54]. Coffee consumption has also been shown to be beneficial in patients who had early stage colon cancer. Coffee consumption has been shown to decrease mortality risk, even after adjusting for other potential confounders such as glycemic index, physical activity and other dietary factors [55]. In another observational study, higher coffee intake in patients who had Stage III Colon cancer was associated with a decreased CRC-specific and all-cause mortality rate [56]. Each cup of coffee translated to 18% lower CRC-specific mortality rate and 20% lower all-cause mortality rate [56]. In a different observational study, patients who increased dietary fiber intake after being diagnosed with colorectal cancer had lower CRC specific and all-cause mortality (19% and 14% lower risk respectively for every 5 g/day increase) [46]. Diets rich in nuts was also associated with an increased DFS and Over-all survival. A recent study among patients with Stage III Colorectal cancer showed that higher tree nut consumption and healthy lifestyle was associated with reduced cancer recurrence by 42% and reduced mortality by 57% [57]. While obesity is known to increase the risk of obesity related cancers such as renal cancer, pancreatic cancer, postmenopausal breast cancer, esophageal, endometrial cancer, it unclear if weight loss in CRC survivors improves outcomes in long term survival [53]. Higher levels of exercise in CRC survivors have also shown to improve CRC specific mortality, all-cause mortality [58] and improving fatigue, quality of living and functional status. 

Aspirin and NSAIDS such as celecoxib, through inhibition of COX-2 pathway, has shown to be beneficial in CRC survivors, with a 29% decrease in CRC specific mortality and a 21% decrease in overall mortality [59,60]. However, it is unclear if all patients derive benefit from aspirin use. There are ongoing placebo controlled clinical trials to further evaluate the benefit of aspirin in patients who have received surgical and medical treatment for colorectal cancer. A randomized clinical trial showed that 600mg of Aspirin a day for 25 months significantly decreased cancer incidence in carriers of Lynch Syndrome [61]. Though not endorsed by major organizations, it is generally recommended that CRC survivors be started on aspirin, in the absence of contraindications such as coagulopathy, bleeding episodes, gastritis or peptic ulcer disease. 

Though an association between Vitamin D status and prognosis is often suggested, it is not clear if repleting low Vitamin D levels leads to an improvement in the outcomes of the patients. While observational studies have shown low Vitamin D levels being associated with poor outcomes such as low over-all survival [53,62,63], confounding factors such as performance status of the patient were not taken into account (patients with advanced disease, with poor performance status living indoor, leading to low Vitamin D levels). However, given the benefit of Vitamin D to bone health, it is reasonable to replenish low Vitamin D levels. 

## 8. Conclusions

Though the incidence and the mortality from colorectal cancer has decreased over the last few decades, epidemiological studies suggest that the incidence of colorectal cancer would increase in people less than 50 years of age. Dietary intake of a person, nutritional status, physical activity and other changes have shown to be associated with pathogenesis of colorectal cancer and also suggested to be associated with poor outcomes as described above. However, definitive randomized control trials to demonstrate a causal relationship is needed in the near future to define high risk factors that may play a preventive and also a prognostic role in colorectal cancer. 

## Figures and Tables

**Table 1 nutrients-11-00164-t001:** Summary of major epidemiological studies examining the association of Red Meat Intake with CRC.

Author, Year	Study Design	Country/Center	Result
Chao A, 2005 [7]	Observational	USA	Positive association—long term meat consumption increased the risk of cancer in the distal portion of the large intestine
Norat A, 2005 [8]	Case-Control	10 European Countries	Positive association—high intake (>160 g/day) group had a risk 1.35-fold as compared with the lowest intake (<20 g/day)
Willett WC, 1990 [9]	Observational	USA	Positive association - RR of CRC in women who ate beef, pork or lamb as a main dish every day was 2.49, as compared with those reporting consumptions less than once a month.
Cross AJ, 2010 [10]	Observational	USA	Positive association—heme iron, nitrite, heterocyclic amines from meat may explain these associations
Chan DS, 2011 [11]	Meta-analysis of ten cohort studies	N/A	Positive association—17% increased risk per 100 g per day of red meat and an 18% increase per 50 g per day of processed meat
Beresford SA, 2006 [12]	Randomized controlled trial	USA	No association—a low-fat dietary pattern intervention did not reduce the risk of colorectal cancer in postmenopausal women during 8.1 years of follow-up

**Table 2 nutrients-11-00164-t002:** Summary of major epidemiological studies examining the association of Alcohol consumption with CRC.

Author, Year	Study Design	Country/Center	Result
Fedirko, 2011 [16]	Meta-analysis of 27 cohort and 34 case-control studies	USA, Europe, Asia, Australia	Positive—strong evidence for an association between alcohol drinking of >1 drink/day and CRC risk.
Cho E, 2004 [17]	Pooled analysis of 8 cohort studies	North America and Europe	Positive—Increased risk for CRC was limited to persons with an alcohol intake of 30 g/d or greater (approximately >or =2 drinks/d)
Mizoue T, 2008 [18]	Pooled analysis of 5 cohort studies	Japan	Positive—Increased risk with an alcohol intake of >or =23 g/day.

**Table 3 nutrients-11-00164-t003:** Summary of major epidemiological studies examining the association of Vitamin D intake with CRC.

Author, Year	Study Design	Country/Center	Result
McCullough, 2018 [21]	Pooled Analysis of 17 cohorts	USA, Europe	Inverse relationship
Chung M, 2011 [22]	Systematic Review	USA	Inverse relationship

**Table 4 nutrients-11-00164-t004:** Summary of major epidemiological studies examining the association of Calcium intake with CRC.

Author, Year	Study Design	Country/Center	Result
Shaukat A, 2005 [26]	Meta-analysis	USA	Recurrence of adenomas was significantly lower in subjects randomized to calcium supplementation
Zhang X, 2016 [27]	Observational	USA	Inverse relationship
Wactawski-Wende, 2006 [25]	Randomized controlled trial	USA	Daily supplementation of calcium (1000 mg of elemental Ca) with vitamin D (400 IU of Vit D3) for 7 years had no effect on the incidence of CRC among postmenopausal women
Lappe J, 2017 [24]	Randomized Controlled Trial	USA	Among healthy postmenopausal older women with a mean baseline serum 25-hydroxyvitamin D level of 32.8 ng/mL, supplementation with vitamin D3 (2000 IU/day) and calcium (1500 mg/day) compared with placebo did not result in a significantly lower risk of all-type cancer at 4 years

**Table 5 nutrients-11-00164-t005:** Summary of major epidemiological studies examining the association of fiber intake with CRC.

Author, Year	Study Design	Country/Center	Result
Peters U, 2003 [34]	Observational	USA	Dietary fiber, particularly from grains, cereals and fruits, was associated with decreased risk of distal colon adenoma
Bingham, 2003 [32]	Observational	Europe	Inverse relationship; In populations with low average intake of dietary fiber, an approximate doubling of total fiber intake from foods could reduce the risk of colorectal cancer by 40%.
Larsson SC, 2005 [33]	Observational	Sweden	Inverse relationship-high consumption of whole grains may decrease the risk of colon cancer in women
Dahm CC, 2010 [47]	Case-Control	UK	Inverse relationship
Fuchs CS, 1999 [36]	Observational	USA	No protective effect of fiber
Beresford SA, 2006 [12]	Randomized Controlled Trial	USA	No protective effect
Park Y, 2005 [40]	Pooled Analysis	USA, Europe	After accounting for other dietary risk factors, high dietary fiber intake was not associated with a reduced risk of colorectal cancer.
MacLennan R, 1995 [37]	Randomized Controlled Trial	Australia	No significant prevention of new adenomas
Schatzkin A, 2000 [38]	Randomized Controlled Trial	USA	Low fat and high fiber diet including fruits and vegetables did not influence the risk of recurrence of colorectal adenomas.
Alberts DS, 2000 [39]	Randomized Controlled Trial	USA	Wheat bran fiber did not protect against colorectal adenomas.
Asano T, 2002 [48]	Meta-analysis	USA, Canada	No evidence from RCTs to suggest that increased dietary fiber intake will reduce the incidence or recurrence of adenomatous polyps within a two to four year period.
Mehta RS, 2017 [42]	Prospective Cohort Study	USA	Prudent diets rich in whole grains and dietary fiber were associated with a lower risk for F. nucleatum-positive colorectal cancer but not F. nucleatum- negative cancer

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
