# Peer review of "Colorectal Cancer and Nutrition"

_nutrients, 2019, doi:10.3390/nu11010164_

Round 1
Reviewer 1 Report
Nutrients
Manuscript ID: nutrients-405362
Title: Colorectal Cancer - A comprehensive review
Kannan Thanikachalam, Gazala Khan
The present review/short write up on colorectal cancer has focused the role of nutrition on the development of colon cancer. Nutrition is playing both a casual and protective role.
Pros
Apart from the environmental and genetic factors, the article summarizes role of nutrition in the development of colon cancer. Vitamin D has been elaborated in the write-up. Alcohol has been discussed.
Cons
CRC is known to be effected by food habits and the "comprehensive" in the title does not justifies the write up. This is mostly the statistics re-write-up from the Siegel et al data. The review is not focused on different nutrients and their role in CRC. The review has not touched "microbiome" at all, which varies with the type of nutrition.
Author Response
CRC is known to be effected by food habits and the "comprehensive" in the title does not justifies the write up.
Response 1: We realise that "comprehensive" in the title , doesn't justify the article. Since we are just focusing on the dietary factors in the pathogenesis of colorectal cancer, we have changed the title of the article to "Colorectal Cancer - Role of Nutrition in colorectal cancer". We have also included new tables summarising pivotal clinical trials that were performed which demonstrated a relationship to nutrients and colorectal cancer.
2. The review has not touched "microbiome" at all, which varies with the type of nutrition.
Response 2: We have added the role of gut microbiota and the effect of fiber.
3. This is mostly the statistics re-write-up from the Siegel et al data
Response 3: We had taken statistical analysis from Siegel et al, since it provided us the most complete and relevant clinical information that we needed to write this article.
Reviewer 2 Report
The review manuscript focuses on role of nutrition in colorectal cancer (CRC). CRC is one of the leading cause of cancer related death in the world. The authors discussed in details about the risk factors associated with CRC with particular focus on nutrients. However, several key areas of CRC such as classification, clinical presentation, location, staging, latest treatment options available were mentioned as a passing comments in the review. The title does not reflect the body of the review manuscript. References are missing in many areas of the manuscript and 29 references are not adequate for a review article. Since this is comprehensive review the authors need to summarize details of the studies that have been done related to nutrients in CRC in a table form.
At this stage the review manuscript lacks the quality of a comprehensive review.
Author Response
1. The review manuscript focuses on role of nutrition in colorectal cancer (CRC). CRC is one of the leading cause of cancer related death in the world. The authors discussed in details about the risk factors associated with CRC with particular focus on nutrients. However, several key areas of CRC such as classification, clinical presentation, location, staging, latest treatment options available were mentioned as a passing comments in the review. The title does not reflect the body of the review manuscript.
Response 1: We realise that "comprehensive" in the title , doesn't justify the article. Since our intent in writing this article was to mainly focus on the dietary factors in the pathogenesis of colorectal cancer, we have changed the title of the article to "Colorectal Cancer - Role of Nutrition in colorectal cancer". We did not include detailed information about location , staging , clinical presentation or latest treatment options in this article since it was beyond the scope of this article.
2. References are missing in many areas of the manuscript and 29 references are not adequate for a review article
Response 2: We have added many more references to this article, a total of 64 .
3. Since this is comprehensive review the authors need to summarize details of the studies that have been done related to nutrients in CRC in a table form.
Response 3: We have summarized pivotal clinical trials that demonstrated an relation to nutrients in CRC.
Round 2
Reviewer 2 Report
In the updated version of the review manuscript titled "Colorectal Cancer - A comprehensive review" the authors have taken into accounts the reviewers' comments and incorporated the recommended corrections. The review manuscript would be an important addition to the journal.
The authors are recommended to change the updated titled “Colorectal cancer: Role of nutrition in colorectal cancer” as there are redundant words.
Suggested title, “Role of nutrition in colorectal cancer” Or “Colorectal cancer and nutrition”
Best of luck.
Author Response
Response:
As recommended by the reviewer, we have changed the title of the updated manuscript to "Colorectal Cancer and Nutrition"